# Association of inflammation-related markers and diabetic retinopathy severity in the aqueous humor, but not serum of type 2 diabetic patients

Lucia Saucedo[1], Isabel B. Pfister[1], Christin Schild[1], Justus G. Garweg[1,2]*

**1** Swiss Eye Institute, Rotkreuz, and Berner Augenklinik, Bern, Switzerland, **2** Department Ophthalmology, Inselspital, University of Bern, Bern, Switzerland

* justus.garweg@augenklinik-bern.ch

**Data Availability Statement:** All relevant data are within the paper and its Supporting Information files.

## Abstract

Diabetic retinopathy (DR) is a frequent microvascular complication of diabetes mellitus, and inflammatory pathways have been linked to its pathogenesis. In this retrospective, observational pilot study, we aimed to compare the concentrations of four inflammation-related proteins, ZAG, Reg-3a, elafin and RBP-4, in the serum and aqueous humor of healthy controls and diabetic patients with different stages of DR. The concentrations of VEGF-A, IL-8, IL-6 were determined in parallel as internal controls. In the serum, we did not find significant differences in the concentrations of target proteins. In the aqueous humor, higher levels of ZAG, RBP-4, Reg-3a and elafin were observed in advanced nonproliferative DR (NPDR)/ proliferative DR (PDR) compared to controls. The levels of ZAG and RBP-4 were also higher in advanced NPDR/PDR than in nonapparent DR. Normalization of target protein concentrations to the aqueous humor total protein demonstrates that a spill-over from serum due to breakage of the blood-retina barrier only partially accounts for increased inflammation related markers in later stages. In conclusion, we found elevated levels of Reg-3a, RBP-4, elafin and ZAG in advanced stages of diabetic retinopathy. Higher levels of pro-inflammatory proteins, Reg-3a and RBP-4, might contribute to the pathogenesis of diabetic retinopathy, as the parallel increased concentrations of anti-inflammatory molecules elafin and ZAG might indicate a compensatory mechanism.

## Introduction

Diabetic retinopathy (DR) is a common microvascular complication of diabetes mellitus (DM), affecting approximately 13% of patients with newly diagnosed DM type 2 (T2DM) [1] and 26% of diabetic patients in Europe [2]. Inflammation is a key mechanism in the pathogenesis of diabetic retinopathy. DM starts early to cause chronic low-grade systemic and local inflammation, which leads to leukostasis, capillary occlusion, and hypoxia. The latter causes endothelial damage and impairment of the blood–retina barrier (BRB), which ultimately leads

**Funding:** This research was funded by the Stiftung Lindenhof Bern (grant number 17-05-F). The funders had no role in study design, data collection and analysis, decision to publish, or preparation of the manuscript.

**Competing interests:** Justus G. Garweg acts as an advisor and speaker for several pharmaceutical companies (AbbVie, Roche, Bayer, and Novartis) and participates in several international industrysponsored clinical studies. This does not alter our adherence to PLOS ONE policies on sharing data and materials. All the authors have declared that no competing interests exist.

to microaneurysm formation, hemorrhages, exudates and diabetic macular edema (DME) [3]. Activated pericytes, endothelial cells and microglia are the major sources of proinflammatory factors, including vascular endothelial growth factor (VEGF), a key regulator of physiological and pathological angiogenesis. Diabetes-driven pathological neovessel formation is the hallmark of proliferative DR (PDR), which includes the formation of new, unstable retinal blood vessels in response to hypoxia, which can lead to vitreous hemorrhage, tractional retinal detachment, and severe vision loss [4].

Several inflammatory mediators have been identified in the serum, aqueous humor (AH) and vitreous fluid (VF) of DR patients, including pro- and anti-inflammatory cytokines and chemokines, angiogenic and anti-angiogenic growth factors, and adhesion molecules and integrins, among others [5, 6]. Some of these have already been discussed as potential biomarkers of DR severity and/or potential drug targets [7–10]. Inflammation is a known key driver in the pathogenesis of DR, enforcing the search for potential new and confirmation of reported biomarkers to further elucidate the complex pathophysiological mechanisms namely behind the advance from early to late stages in diabetic retinal pathologies.

Four inflammation-related proteins involved in DR have been evaluated as potential biomarkers of ocular disease. One of these is Retinal Binding Protein 4 (RBP-4), which is secreted by hepatocytes and adipocytes and is responsible for the transport of retinol from the liver to target tissues, such as adipose tissue, the retina and the brain [11]. High levels of RBP-4 have been associated with metabolic disease and insulin resistance [12]. Furthermore, RBP-4 has been shown to induce the secretion of proinflammatory cytokines in macrophages and therefore has been proposed as a biomarker of systemic and local inflammation and DR [11, 13]. Regenerating islet-derived protein 3 alpha (Reg-3a) is a member of the Reg family of proteins. Reg proteins have also been implicated in various pathologies, such as Alzheimer's disease, cancer and diabetes [14]. Reg proteins have antiapoptotic, trophic, angiogenic and anti-inflammatory functions [15], and elevated levels have been reported in several inflammatory diseases, such as Crohn´s disease and pancreatitis [16]. A third inflammation-related protein is elafin, a human elastase-specific protease inhibitor with antimicrobial activity and that is primarily expressed in immune cells, the intestinal tract, vagina, lungs, and skin. Elafin clearly regulates the inflammatory response in monocytes and endothelial cells by inhibiting NF-kB activation [17, 18]. Finally, the role of zinc-α2-glycoprotein (ZAG), a member of the adipokine family involved in lipid and glucose metabolism, has controversially been discussed [19]. It has been associated with insulin resistance and has anti-inflammatory properties [20]. ZAG overexpression appears to mitigate inflammatory responses and improve mitochondrial function [21].

In this pilot study, we aimed to determine the concentrations of the inflammation-related proteins RBP-4, Reg-3a, elafin and ZAG in the serum and AH of healthy controls and diabetic patients with and without different stages of DR to explore their potential use as biomarkers of DR.

## Materials and methods

### Patients

This retrospective study drew on 42 patients with T2DM with or without DR and 19 healthy individuals without any known systemic or ocular disease who underwent cataract surgery. Serum and AH samples were obtained at the beginning of the surgery in the Clinic for Vitreoretinal Diseases, Berner Augenklinik, Bern, Switzerland. Exclusion criteria included any other ocular disease except cataract and DR, i.e., vitreous hemorrhage, uveitis, glaucoma, or any concomitant retinal pathology history and any systemic malignant, vascular or inflammatory comorbidity (e.g., rheumatic or autoimmune diseases), systemic treatment including

corticosteroids or immunomodulatory drugs, intravitreal therapy, panretinal laser photocoagulation within 6 months prior to surgery, and any ocular treatment except lubricants.

The stage of DR was determined by a graduated ophthalmologist blinded to the study protocol based on the results of dilated stereo biomicroscopy of the anterior and posterior segments of the eye, macular optical coherence tomography (OCT) and wide field fundus images (Optos®) according to the International Clinical Diabetic Retinopathy Disease Severity Scale [22]. Ocular disease was categorized as diabetes with nonapparent DR (n = 20), mild to moderate nonproliferative DR (NPDR) (n = 10), and advanced NPDR/PDR (n = 12).

Serum and AH samples were obtained between 2015 and 2022 from patients at the beginning of cataract surgery and immediately stored at -80°C until analysis of all samples in parallel, which was conducted between June and July 2022. The patients' clinical data were obtained from their medical history and coded afterwards for analysis. The patients included in this study attended and had their follow up in the Clinic for Vitreoretinal Diseases, Berner Augenklinik, Bern, Switzerland and had provided prior to the surgery written general consent to the sampling and analysis of their biological materials and clinical data. This study was approved by the local Ethics Committee of the University of Bern (Ref. no 2019–00651) and performed in full compliance with the tenets of the Declaration of Helsinki. All samples were analyzed using multiplex immunoassay kits based on Luminex MAP technology, as previously described [23] following the manufacturer's instructions. Elafin, ZAG, Reg-3a and RBP-4 were determined using the Abundant Serum Markers 26-Plex Human Panel assay (Thermo Fisher Scientific, Waltham, Massachusetts, USA). Serum and AH levels of VEGF, IL-6, and IL-8 were determined in parallel as an internal quality control system using a ProCarta-Plex customized assay (ThermoFisher Scientific, Waltham, Massachusetts, USA). For the 26-plex assay, serum samples were prediluted 1:100, and AH samples were prediluted 1:10 in assay buffer, as per the manufacturer's instructions. No predilution was used for the ProCarta-Plex customized assay. The plates were read using the Bio-Plex Flexmap 3D system with xPONENT 4.2 software (Bio-Rad, Hercules, CA, USA). All samples were analyzed in parallel and in a blinded manner to avoid bias.

The following clinical variables have been measured: age (years), gender, body mass index (BMI) (calculated as weight (kg)/ [height (m)]$^2$), duration of diabetes (in years), hypertension, and medication intake. Target inflammation-related proteins have been measured as reported above. Since this is a pilot study including only Swiss patients we cannot exclude any kind of bias. To minimize impact of bias we included both male and female patients as far as available, as well as different stages of retinopathy. Further, we controlled for the impact of a variety of medications. Two researchers extracted the clinical data and did a double coding to avoid a rater bias. The proteomic analyses were done by a technician without knowledge of the grouping and underlying diagnoses for the samples.

The total protein concentration of serum and AH from all patients was quantified simultaneously using the Coomasie (Bradford) Protein Assay kit (Thermo Fisher Scientific, Waltham, Massachusetts, USA) according to the manufacturer's instructions. Briefly, 5 μl of AH or prediluted serum (1:200) was placed in a 96-plex plate with 250 μl of Coomassie reagent and incubated for 10 min. Absorbance was read at 595 nm.

## Statistical analysis

The Shapiro–Wilk test was applied to determine the normal distribution of the data. Outliers were identified by a box-plot analysis (box-whisker plot), and extreme outliers (more than three box lengths away from the edge) were excluded from the statistical analysis.

In nonnormally distributed data, the nonparametric Kruskal–Wallis H test was used for intergroup comparison of continuous data, and the chi-square test of independence was used to evaluate variables measured at a nominal level. A $p < 0.05$ was set as significant. To control the risk of introducing type I error as a result of multiple comparisons, we applied the Holm correction, which progressively adapts the threshold for rejecting the null hypothesis. For normalization of the AH target protein concentration, the AH concentration of each target protein was further normalized to the total AH protein concentration of each sample. An average of all the normalized values was calculated for each group, and the nonparametric Kruskal–Wallis H test followed by Holm's correction was applied. All statistical analyses were performed using the open-source software R (Version 3.3.2±2016 RStudio, Inc.; psych package) and SPSS (version 23.0; IBM SPSS Statistics, Armonk, NY, USA).

## Results

### Demographic data and medication

Demographic data and systemic medication for the patients are presented in Table 1. Study groups presented no differences in age or frequencies of hypertension, statin, fibrate and sartane use. Significant differences in gender were observed in the mild/moderate NPDR group and in the advanced NPDR/PDR group in comparison to control patients. Significant differences in BMI in the nonapparent DR group in comparison to the control group were also observed. Regarding the diabetic groups, patients with advanced NPDR/PDR presented differences in the duration of diabetes and insulin and metformin intake in comparison to the advanced DR groups.

### Inflammatory markers in the serum

Serum levels of ZAG, Reg-3a, RBP-4 and elafin were measured using a multiplex assay, as were those of VEGF-A, IL-6 and IL-8, which were used as internal quality controls. Serum RBP-4 and IL-6 ranged within the limits of detection of the assay in only 34% and 13% of samples, respectively, and were thus excluded from the statistical analyses. All the other target proteins ranged within the limits of detection of the assay in all samples.

The results of multiplex analysis (Fig 1) did not reveal relevant differences in the serum concentrations of the target biomarkers, since statistically significant differences were no longer significant after Holm's correction. A tendency of higher ZAG levels in patients with advanced NPDR/PDR compared to mild/moderate NPDR patients ($p = 0.027$) was observed. Serum from patients with nonapparent DR tended to present higher levels of Reg-3a than healthy controls ($p = 0.008$), and serum from patients with advanced NPDR/PDR presented higher Reg-3a levels than controls ($p = 0.008$). Additionally, a tendency toward higher elafin levels was observed in patients with advanced NPDR/PDR compared to healthy controls, diabetic patients with no apparent DR and patients with mild/moderate NPDR ($p = 0.046$, 0.04 and 0.027, respectively). As expected, serum concentrations of VEGF-A and IL-8 did not reveal significant differences between the groups.

### Inflammatory markers in the aqueous humor

All target proteins ranged within the detection limits of the assay in all AH samples. Higher AH levels of ZAG and RBP-4 in advanced NPDR/PDR compared to healthy controls ($p = 0.0011$ and $p = 0.00016$, respectively) as well as in advanced NPDR/PDR compared to nonapparent DR patients were found ($p = 0.000077$ and $p = 0.00067$, respectively). Furthermore, elevated levels of Reg-3a were observed in patients with advanced NPDR/PDR

**Table 1. Patient baseline characteristics.**

| | Healthy controls (n = 19) | No apparent DR (n = 20) | Mild/moderate DR (n = 10) | Advanced NPDR/PDR (n = 12) | p value |
|---|---|---|---|---|---|
| Age (years; mean ± SD) | 72.4 ± 8.5 | 70.4 ± 9.9 | 73.7 ± 6.5 | 68.4 ± 8.8 | 0.38 |
| Gender F n (%) | 12 (63.2) | 10 (50.0) | 1 (10.0) | 1 (8.3) | 0.003 |
| BMI (mean ± SD) | 23.5 ± 3.6 | 31.7 ± 8.0 | 30.0 ± 4.2 | 29.2 ± 6.9 | 0.021 |
| Duration of diabetes (years; mean ± SD) | N/A | 17.0 ± 7.4 | 17.5 ± 3.5 | 35.7 ± 13.2 | 0.017 |
| Hypertension n (%) | 0 (0) | 12 (60) | 6 (60) | 7 (58) | 0.71 |
| **Medication (n, %)** | | | | | |
| Insulin | 0 (0.0) | 5 (25) | 4 (40) | 10 (83.3) | 0.006 |
| Metformin | 0 (0.0) | 16 (80) | 6 (60) | 2 (16.7) | 0.001 |
| Statin | 0 (0.0) | 7 (35) | 4 (40) | 4 (33.3) | 0.71 |
| Fibrate | 0 (0.0) | 0 (0) | 2 (20) | 0 (0) | N/A |
| Sartane | 0 (0.0) | 12 (60) | 5 (50) | 9 (75) | 0.68 |

DR: diabetic retinopathy; NPDR: nonproliferative DR; PDR: proliferative diabetic retinopathy; n: sample size; SD: standard deviation; F: female; BMI: body mass index. N/A: not applicable.

compared to healthy controls and to patients with mild/moderate NPDR ($p = 0.00022$ and $p = 0.0021$, respectively). Reg-3a levels tended to be higher in nonapparent DR patients than in healthy controls ($p = 0.008$) and in advanced NPDR/PDR patients than in nonapparent DR patients ($p = 0.008$). Increased elafin concentrations were observed in the advanced NPDR/PDR group in comparison to healthy controls ($p = 0.00087$). In addition, elafin levels tended to be higher in advanced NPDR/PDR than in nonapparent DR and mild/moderate NPDR ($p = 0.027$ in both cases) (Fig 2).

Confirming the assay performance, higher levels of VEGF-A, IL-8 and IL-6 were observed in advanced NPDR/PDR patients compared to healthy controls ($p = 3.2 \times 10^{-5}$, $p = 0.0005$ and $p = 0.0015$, respectively). VEGF-A levels were further increased in advanced NPDR/PDR compared to nonapparent DR ($p = 0.0001$). Finally, higher VEGF-A and IL-6 levels were observed in advanced NPDR/PDR compared to mild/moderate NPDR ($p = 0.0023$ and $p = 0.0021$, respectively) (Fig 3). Concentrations of IL-8 and IL-6 tended to be higher in advanced NPDR/PDR than in nonapparent DR ($p = 0.017$ and $p = 0.027$, respectively; Fig 3).

## Normalization of AH proteins to total AH protein concentration

The total protein concentration in serum and AH was determined using the Bradford assay. The mean levels for each group are displayed in Table 2. Total serum protein levels were comparable throughout all study groups. Total protein levels in the AH were increased in advanced NPDR/PDR compared to healthy controls. Consequently, the AH/serum total protein quotient differed between advanced NPDR/PDR and nonapparent DR.

Given the expected increase in the total protein concentration in the AH in the advanced NPDR/PDR group, the higher levels of the target proteins described previously might be attributed to increased local production or to breakdown of the BRB. To determine the impact

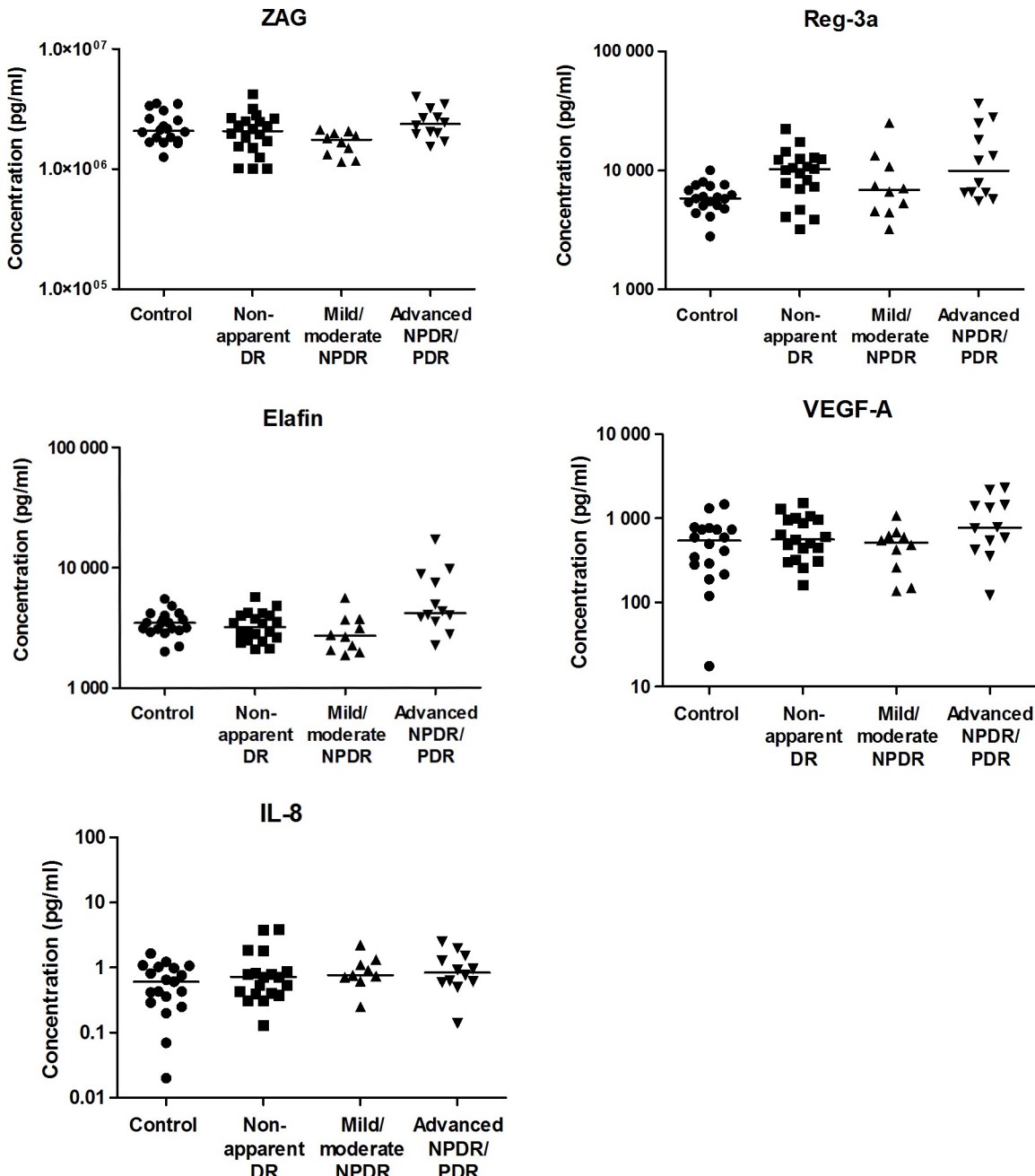

**Fig 1. Serum concentrations of ZAG, Reg-3a, elafin, VEGF-A and IL-8 in healthy controls and diabetic patients with and without different severities of DR.** Each point represents data from a single patient (log scale pg/ml). The black lines show the median value.

of damage to the BRB, the value of each target protein in the AH was normalized to the total AH protein concentration (Fig 4). Differences between study groups for ZAG, RBP-4, Reg-3a and IL-6 (Table 3) lost their significance after applying Holm's correction. No differences between groups in VEGF-A and IL-8 levels were observed (Fig 5).

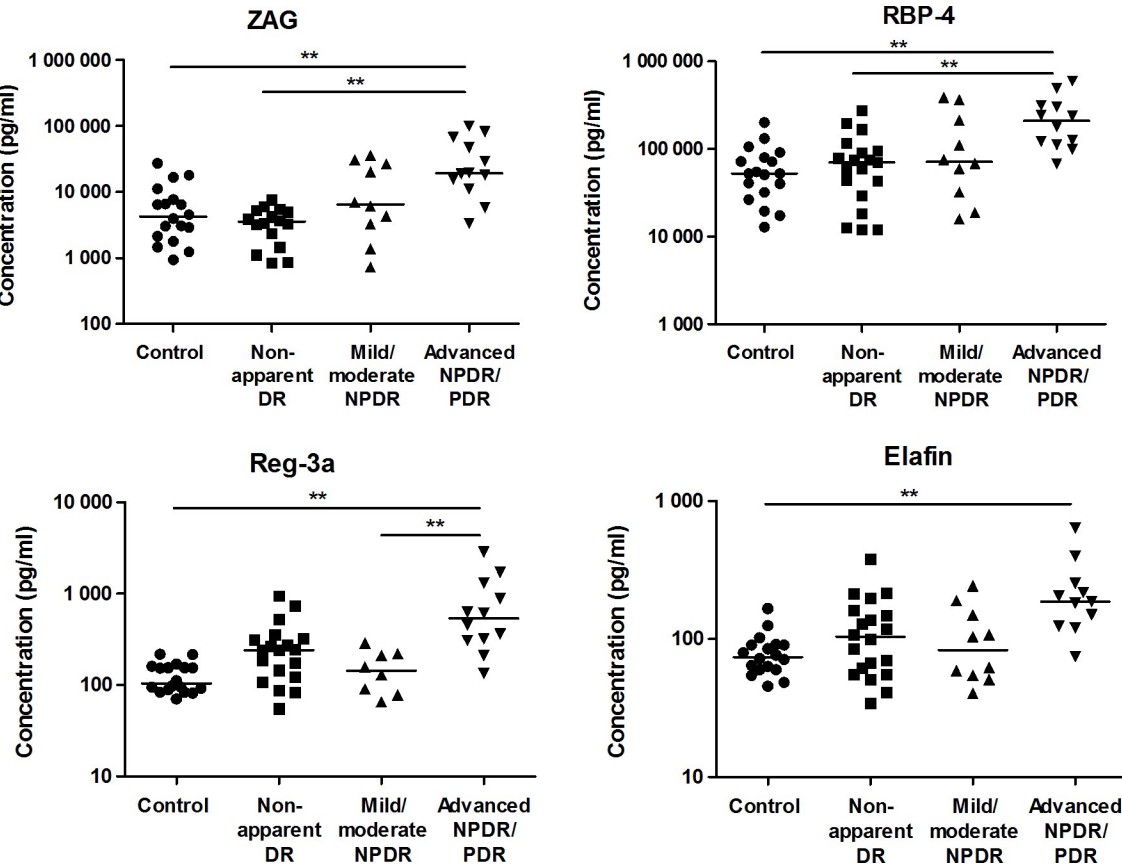

**Fig 2. Aqueous humor concentrations of ZAG, RBP-4, Reg-3a and elafin in healthy controls and diabetic patients with or without different levels of DR.** Each point represents data from a single patient (log scale pg/ml). The black lines show the median value. **$p < 0.01$.

## Discussion

In the underlying pilot study, we report elevated AH levels of ZAG, RBP-4, Reg-3a and elafin in patients with advanced NPDR/PDR in comparison to healthy controls, suggesting their involvement in the pathogenesis of DR and their potential use as biomarkers of late disease.

Elevated serum levels of Reg-1a (another member of the Reg family) have already been reported in patients with T2DM, and they have been linked to diabetes duration and severity [24]. Consistent with this interpretation, Reg-1a has also been proposed as a marker of diabetic nephropathy [25]. In line with these findings, we observed elevated AH Reg-3a levels in later stages of DR. Increased Reg-3a expression seems to be a less specific finding, which has been reported in several diseases with underlying inflammatory pathophysiology. Thus, Reg-3a may act as a cellular response marker of inflammatory damage [16]. Moreover, a neuroprotective effect of Reg-3a was observed *in vitro* and in an *in vivo* model of excitotoxicity [26]. On a systemic level, this recognition has already been applied to treatment protocols with glucagon-like peptide-1 (GLP-1)-based therapies. GLP-1 receptor agonists (GLP-1Ras) and dipeptidyl peptidase-4 inhibitors (DPP-4Is) are commonly used for the treatment of T2DM [27]. Since endocitotoxicity has more recently been identified as a key pathological mechanism underlying DR [28–31], systemic neuroprotective therapies for extraocular disease are likely to indirectly provide benefits for ocular disease, although this has yet to be shown [32, 33]. Based on this robust pathophysiological background with respect to the role of Reg proteins in

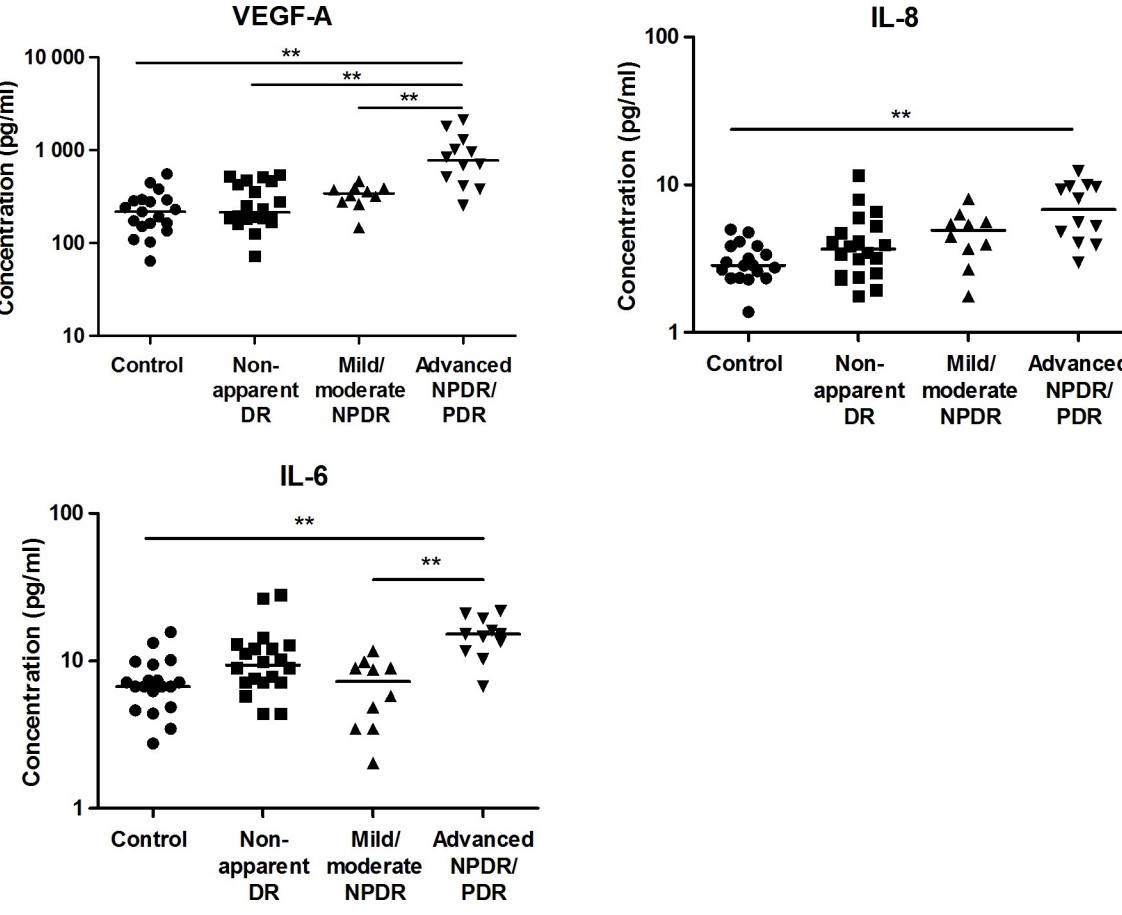

**Fig 3. Aqueous humor concentrations of VEGF-A, IL-8 and IL-6 in healthy controls and diabetic patients with or without different levels of DR.** Each point represents data from a single patient (log scale pg/ml). The black lines show the median value. **$p < 0.01$.

excitotoxicity and neuroprotection in systemic disease, we were surprised that their potential role in the diabetic retina has not previously been reported. Interestingly, we found based on a limited sample size some support for the local production of Reg-3a in eyes with nonapparent DRP, the cellular source of which has yet to be established.

The adipokine RBP-4 has been proposed as a systemic inflammatory biomarker in DR [13], and an increase in systemic RBP-4 in DR has been shown [34–36], but its role at the local level has not been assessed. In the present study, we could not quantify and thus were not able to confirm increased serum RBP-4 in advanced DR. Given the multiplex nature of the analysis,

**Table 2. Total protein concentration (mean ± SD; µg/ml).**

|  | Healthy controls | Nonapparent DR | Mild/moderate DR | Advanced NPDR/PDR | p value |
|---|---|---|---|---|---|
| Serum (x10³) | 90.24 ± 24.0 | 103.9 ± 26.7 | 72.8 ± 27.1 | 98.8 ± 21.4 | 0.099 |
| AH | 226.2 ± 142.1 | 216.2 ± 122.1 | 329.0 ± 206.1 | 445.9 ± 215.7 | 0.018* |
| AH/Serum quotient | 0.0023 | 0.0024 | 0.0035 | 0.0047 | 0.029** |

* difference between controls and advanced NPDR/PDR

** difference between nonapparent DR and advanced NPDR/PDR

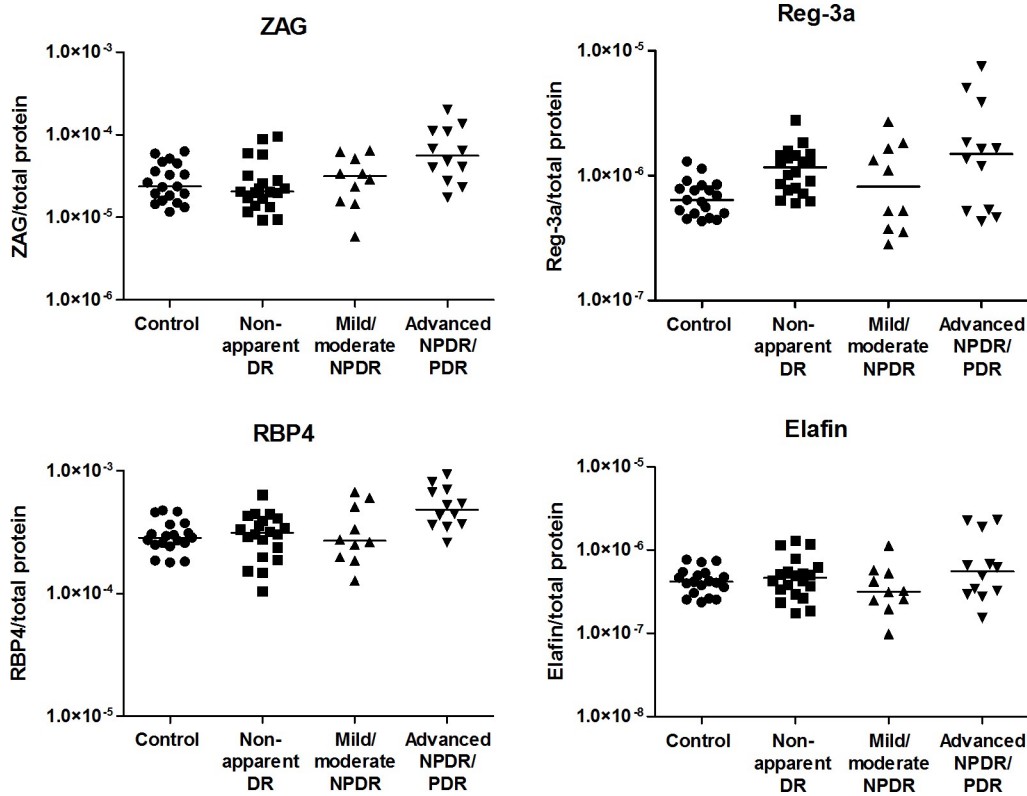

**Fig 4. Aqueous humor concentrations of ZAG, RBP-4, Reg-3a and elafin normalized to AH total protein concentration in healthy controls and diabetic patients with or without different levels of DR.** Each point represents data from a single patient (log scale pg/ml). The black lines show the median value.

serum samples had to be prediluted to suit the detection of most analytes. This predilution, however, did not fit the expected serum concentration range for RBP-4, which thus fell above the detection level. Since serum RBP-4 concentrations have been linked to the severity of DR [37], it is surprising that the RBP-4 concentration in the AH of eyes with different stages of DR has never been reported. On the other hand, RBP-4 is mainly secreted by hepatocytes and adipose tissue, both of which are not found within the eye [12]. Our finding of increased RBP-4 in the AH in advanced NPDR/PDR suggests its contribution to the progression of DR. *In vitro*, RBP-4 induces the expression of proinflammatory cytokines, such as IL-6 and ICAM-1, in endothelial cells and increases leukocyte adherence [38], which is consistent with the current model. Interestingly, overexpression of RBP-4 *in vivo* in the mouse model causes early

**Table 3. Group comparisons of ZAG, RBP-4, Reg-3a and IL-6 concentrations in the AH normalized to AH total protein concentration.**

|         | 1 vs. 2 | 1 vs. 3 | 1 vs. 4 | 2 vs. 3 | 2 vs. 4 | 3 vs. 4 |
|---------|---------|---------|---------|---------|---------|---------|
| ZAG     | >0.05   | >0.05   | **0.015** | >0.05 | 0.012   | >0.05   |
| RBP-4   | >0.05   | >0.05   | **0.003** | >0.05 | 0.004   | **0.049** |
| Reg-3a  | **0.003** | >0.05 | >0.05   | >0.05   | >0.05   | >0.05   |
| IL-6    | >0.05   | **0.022** | >0.05 | **0.022** | >0.05 | >0.05   |

1: healthy controls; 2: nonapparent DR; 3: mild/moderate NPDR; 4: advanced NPDR/PDR. The other proteins were not included, as no differences were observed. Given multiple comparisons, the level of significance has to be set to p = 0.0012 after Holm correction.

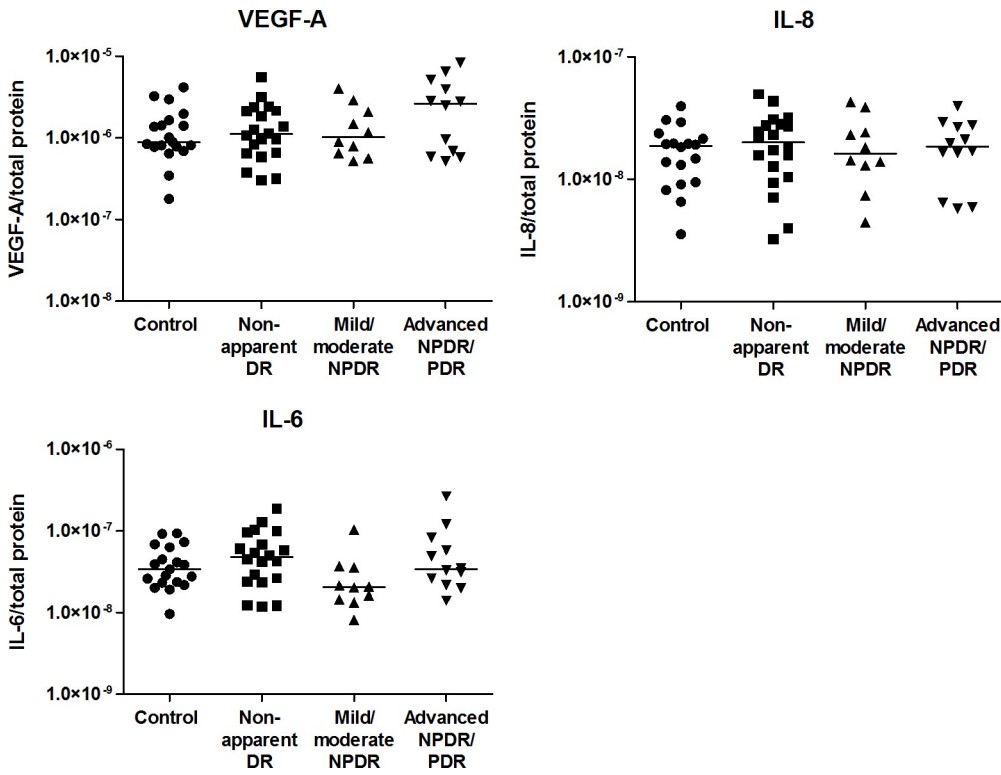

**Fig 5. Aqueous humor concentrations of VEGF-A, IL-8 and IL-6 normalized to AH total protein concentration in healthy controls and diabetic patients with or without different levels of DR.** Each point represents data from a single patient (log scale pg/ml). The black lines show the median value.

microglial activation and increased retinal levels of the proinflammatory cytokine IL-18. The induction and progression of RBP-4-induced retinal neurodegeneration appears to be independent of retinal vascular pathology, obesity, dyslipidemia, and hyperglycemia [39]. Finally, RBP-4 increases mitochondrial superoxide generation and induces endothelial cell apoptosis *in vitro* [40]. In patients with obesity, RBP-4 promotes oxidative stress by decreasing endothelial mitochondrial function. Therefore, it may be a potential biomarker with prognostic potential in obesity and T2DM and could provide a potential therapeutic target on a systemic level for the treatment of these diseases [11]. While its role in DR is supported by its pathophysiological effects, its source within the eye is yet to be established. In this context, we were unable to demonstrate its local production, and spillover from the blood seems likely in nonapparent and early NPDR while a local production in late-stage DR compared to controls is conceivable.

In our series of type 2 diabetics, we observed an increase in AH levels of elafin in advanced NPDR/PDR compared to healthy controls. *In vitro*, elafin plays a protective role against the inflammatory response. Overexpression of elafin reduces IL-8 secretion by endothelial cells and TNF-α secretion by monocytes in a model of induced inflammation *in vivo*, and this effect is mediated by reduced activation of the inflammatory transcription factor NF-kB [18]. In neonatal lungs, elafin overexpression has been shown to suppress inflammation, inhibit apoptosis and decrease MMP-9 activity [41]. Given its protective role in inflammatory responses, this inflammation-related protein could compensate or antagonize other members of the inflammation-related protein family, i.e., RBP-4 and Reg-3a. In the mouse model, elafin seems to increase serum leptin levels and regulate food consumption, thereby inhibiting obesity, hyperglycemia,

and liver steatosis. On the other hand, circulating elafin levels are associated with hyperglycemia in men with T2DM [42], which might be relevant for the interpretation of our results, since a majority of our patients were males with a mean BMI of $\geq$30. Sex differences in diabetic vascular complications are not well understood, and many questions and controversies remain [43]. A closer look into the role of elafin in this context may therefore be justified.

ZAG is a 42 kDa soluble secretory adipokine that has been found in several body fluids. It is thought to play diverse compartment-specific roles, the main one being to promote lipid mobilization [19]. ZAG has been discussed as a biomarker for early kidney disease [44], progression of diabetic nephropathy [45, 46] and kidney disease-associated mortality [47]. Nevertheless, longitudinal data for its role in systemic micro- and macrovascular complications of diabetes have yet to be established. Therefore, elevated levels of ZAG in the AH of our patients with advanced NPDR/PDR might mark advanced microvascular disease. In line with our data, increased vitreal levels of ZAG in patients with PDR in comparison to nondiabetic controls have been reported previously [48]. ZAG is considered an anti-inflammatory adipokine with effects in adipose tissue [49]. In animal models, ZAG overexpression reduces inflammation and prevents hepatic cell apoptosis, while ZAG depletion in a knock-out model has the opposite effect [21]. The finding that ZAG can suppress neuroinflammation mediated by IL-6 and TNF-$\alpha$ [50] may also indicate an interesting aspect for ocular disease. To identify a potential impact of the breakdown of the BRB on the levels of ZAG, RBP-4, Reg-3a and elafin in the AH, we measured the total protein concentration in serum and AH for each patient and normalized the target protein concentration to these values. The AH total protein concentration observed in this study was in line with previously reported values in several diseases [51–53]. Normalization of the target protein concentrations is a standard method to assess local antibody production in viral ocular disease and toxoplasmosis [54] but has rarely been applied to correct for biomarkers, i.e., cytokine concentrations [55]. In this study, we did not observe significant differences among groups after normalization. This suggests that the higher levels of the target protein observed in this study might be derived from serum (as a result of BRB breakdown) rather than from increased local production. Normalization of target proteins to total AH protein levels might provide a better pathophysiological understanding of the origin of increased concentrations of certain proteins observed in late stages of DR. However, it does not provide additional information on the biological relevance of these proteins, since elevated levels of inflammatory markers would still have an impact on the environment in DR regardless of their site of production.

Among the strengths of this study, we included a well-defined DR assessment in all cases and excluded samples of patients with vitreal hemorrhage, intravitreal treatment, or panretinal laser photocoagulation within 6 months before surgery. The limitations include the retrospective nature of our study as well as the limited number of samples. As a result, there was a gender difference in mild/moderate NPDR and advanced NPDR/PDR. Consequently, the results of this pilot study need to be corroborated with a future prospective study including a larger number of patients. Observed differences in BMI are expected, as diabetic patients tend to have higher BMI than healthy subjects [56], and differences in insulin and metformin use are in line with the duration of diabetes and metabolic control in diabetic patients.

Based on our observations, we have initiated a prospective study with more power to further investigate the robustness of these findings and to assess their potential role in type 1 DM. If confirmed, we wish to investigate in a follow up analysis the impact of treatment, i.e. intravitreal anti-VAGF and corticosteroid therapy.

In conclusion, we observed elevated AH levels of Reg-3a, RBP-4, ZAG and elafin in advanced stages of DR. The increase in pro-inflammatory molecules Reg-3a and RBP-4 could contribute to the pathogenesis of DR; in parallel, elevated concentrations of anti-inflammatory

proteins ZAG and elafin might point to a compensatory mechanism attempting to counterbalance this chronic inflammation.

## Supporting information

**S1 Table. Rawdata set.**
(XLSX)

**S1 Checklist. STROBE statement—checklist of items that should be included in reports of observational studies.**
(DOCX)

## Acknowledgments

The authors would like to acknowledge Alain Despont and Robert Rieben for conducting the Bioplex assay and for their support during the development of this work.

## Author Contributions

**Conceptualization:** Lucia Saucedo, Justus G. Garweg.

**Formal analysis:** Lucia Saucedo, Isabel B. Pfister, Justus G. Garweg.

**Funding acquisition:** Justus G. Garweg.

**Investigation:** Lucia Saucedo.

**Project administration:** Christin Schild.

**Supervision:** Justus G. Garweg.

**Visualization:** Lucia Saucedo, Isabel B. Pfister, Christin Schild.

**Writing – original draft:** Lucia Saucedo, Isabel B. Pfister, Justus G. Garweg.

**Writing – review & editing:** Lucia Saucedo, Isabel B. Pfister, Christin Schild,
Justus G. Garweg.

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
