## [Decision Letter · Decision Letter 0]

20 Sep 2023

PONE-D-23-14800Association of Inflammation-related Markers and Diabetic Retinopathy Severity in the Aqueous humor, but not Serum of Type 2 Diabetic PatientsPLOS ONE

Dear Dr. Garweg,

Thank you for submitting your manuscript to PLOS ONE. After careful consideration, we feel that it has merit but does not fully meet PLOS ONE’s publication criteria as it currently stands. Therefore, we invite you to submit a revised version of the manuscript that addresses the points raised during the review process.

The reviews are that this is an excellent manuscript with a few minor issues. 

We look forward to receiving your revised manuscript.

Kind regards,

Andrew W Taylor, Ph.D.

Academic Editor

PLOS ONE

Journal Requirements:

"Justus G. Garweg acts as an advisor and speaker for several pharmaceutical companies (AbbVie, Roche, Bayer, and Novartis) and participates in several international industry-sponsored clinical studies. All the authors have declared that no competing interests exist. "

Additional Editor Comments :

Excellent manuscript. All concerns are minor.

In addition to reviewer 1. On-Line 129, it states that “Two researchers extracted the clinical data and did a double coding to avoid rater bias.” Were the proteomic measurements also done in a blinded manner and then sorted into the groups for statistical analysis? If it was, please include it in the methods.

Reviewers' comments:

Reviewer's Responses to Questions

**Comments to the Author**

1. Is the manuscript technically sound, and do the data support the conclusions?

Reviewer #1: Yes

2. Has the statistical analysis been performed appropriately and rigorously? 

Reviewer #1: Yes

3. Have the authors made all data underlying the findings in their manuscript fully available?

Reviewer #1: Yes

4. Is the manuscript presented in an intelligible fashion and written in standard English?

Reviewer #1: Yes

5. Review Comments to the Author

Reviewer #1: Dear Author

I had the privilege of reviewing your manuscript.

Taking into account the striking difference observed in normal patients versus diabetic patients in the aqueous humor levels of the biomarkers, what are the subsequent studies devised by your group to try to address the increased secretion of those inflammatory biomarkers ?

Do you think intraocular steroid therapy may decrease the secretion of those biomarkers and maybe delay the retinopathy progression?

A new study addressing patients injected with intravitreal slow release dexamethasone may show some promise in controlling that expression?

Those are subjects worth discussing in the manuscript.

Best

6. PLOS authors have the option to publish the peer review history of their article (what does this mean?). If published, this will include your full peer review and any attached files.

Reviewer #1: **Yes: **Rodrigo Antonio Brant Fernandes

---

## [Author Response · Author response to Decision Letter 0]

28 Sep 2023

Pone-D-23-14800 Response to the Editor and reviewer

We thank for the opportunity to revise our paper. The manuscript has been revised and text and tables habe been modified to meet PLOS ONE’s style requirements. Following changes have been made to meet the expectations of editor and reviewer: 

Editor: 

Thank you for stating the following in the Competing Interests section:

"Justus G. Garweg acts as an advisor and speaker for several pharmaceutical companies (AbbVie, Roche, Bayer, and Novartis) and participates in several international industry-sponsored clinical studies. All the authors have declared that no competing interests exist. "

Response:

We have updated the Competing Interests statement and reported it in the cover letter. 

Editor: 

In your Data Availability statement, you have not specified where the minimal data set underlying the results described in your manuscript can be found. PLOS defines a study's minimal data set as the underlying data used to reach the conclusions drawn in the manuscript and any additional data required to replicate the reported study findings in their entirety. All PLOS journals require that the minimal data set be made fully available. For more information about our data policy, please see http://journals.plos.org/plosone/s/data-availability.

Response:

Our data set has been included in a Supporting Information file and is uploaded with the resubmission. 

Editor: 

Please include your full ethics statement in the ‘Methods’ section of your manuscript file. In your statement, please include the full name of the IRB or ethics committee who approved or waived your study, as well as whether or not you obtained informed written or verbal consent. If consent was waived for your study, please include this information in your statement as well.

Response: 

This information can be found in the methods section of the manuscript, lines 107-112

Response: 

The reference list has been revised and no changes were made. No retracted papers have been cited. 

Additional Editor Comments:

Excellent manuscript. All concerns are minor.

In addition to reviewer 1. On-Line 129, it states that “Two researchers extracted the clinical data and did a double coding to avoid rater bias.” Were the proteomic measurements also done in a blinded manner and then sorted into the groups for statistical analysis? If it was, please include it in the methods.

Response: 

We would like to thank the editor for his favourable rating and for the very relevant comment. It was added in the materials and methods section, lines 124-126. 

Reviewers' comments:

Reviewer #1: Dear Author

I had the privilege of reviewing your manuscript.

Taking into account the striking difference observed in normal patients versus diabetic patients in the aqueous humor levels of the biomarkers, what are the subsequent studies devised by your group to try to address the increased secretion of those inflammatory biomarkers ?

Do you think intraocular steroid therapy may decrease the secretion of those biomarkers and maybe delay the retinopathy progression?

A new study addressing patients injected with intravitreal slow release dexamethasone may show some promise in controlling that expression?

Those are subjects worth discussing in the manuscript.

Best

Response: 

We thank very deeply the reviewer for his excellent feedback and for taking the time to review our manuscript. We sincerely appreciate this review. We found the comments very insightful and the raised topics were added in the discussion, lines 381-4: “Based on our observations, we have initiated a prospective study with more power to further investigate the robustness of these findings and to assess their potential role in type 1 DM. If confirmed, we wish to investigate in a follow up analysis the impact of treatment, i.e. intravitreal anti-VAGF and corticosteroid therapy”.

---

## [Editor Report · Decision Letter 1]

6 Oct 2023

Association of Inflammation-related Markers and Diabetic Retinopathy Severity in the Aqueous humor, but not Serum of Type 2 Diabetic Patients

PONE-D-23-14800R1

Dear Dr. Garweg,

We’re pleased to inform you that your manuscript has been judged scientifically suitable for publication and will be formally accepted for publication once it meets all outstanding technical requirements.

Kind regards,

Andrew W Taylor, Ph.D.

Academic Editor

PLOS ONE

Additional Editor Comments (optional):

All issues addressed.
---

## [Editor Report · Acceptance letter]

16 Oct 2023

PONE-D-23-14800R1 

Association of Inflammation-related Markers and Diabetic Retinopathy Severity in the Aqueous humor, but not Serum of Type 2 Diabetic Patients 

Dear Dr. Garweg:

I'm pleased to inform you that your manuscript has been deemed suitable for publication in PLOS ONE. Congratulations! Your manuscript is now with our production department. 

Kind regards, 

on behalf of

Dr. Andrew W Taylor 

Academic Editor

PLOS ONE